# Tunable genetic devices through simultaneous control of transcription and translation

Vittorio Bartoli [1,2], Grace A. Meaker [3], Mario di Bernardo [1,2,4] & Thomas E. Gorochowski [1,5✉]

Synthetic genetic circuits allow us to modify the behavior of living cells. However, changes in environmental conditions and unforeseen interactions with the host cell can cause deviations from a desired function, resulting in the need for time-consuming reassembly to fix these issues. Here, we use a regulatory motif that controls transcription and translation to create genetic devices whose response functions can be dynamically tuned. This allows us, after construction, to shift the on and off states of a sensor by 4.5- and 28-fold, respectively, and modify genetic NOT and NOR logic gates to allow their transitions between states to be varied over a >6-fold range. In all cases, tuning leads to trade-offs in the fold-change and the ability to distinguish cellular states. This work lays the foundation for adaptive genetic circuits that can be tuned after their physical assembly to maintain functionality across diverse environments and design contexts.

[1] BrisSynBio, University of Bristol, Life Sciences Building, Tyndall Avenue, Bristol, UK. [2] Department of Engineering Mathematics, University of Bristol, Woodland Road, Bristol, UK. [3] School of Biosciences, Cardiff University, Museum Avenue, Cardiff, UK. [4] Department of Electrical Engineering and Information Technology, University of Naples Federico II, Via Claudio 21, Napoli, Italy. [5] School of Biological Sciences, University of Bristol, Tyndall Avenue, Bristol, UK. ✉email: thomas.gorochowski@bristol.ac.uk

Genetic regulatory circuits govern when and where genes are expressed in cells and control core biochemical processes such as transcription and translation[1,2]. The ability to synthesize DNA encoding engineered genetic circuits offers a means to expand the capabilities of a cell and reprogram its behavior[1,3]. Synthetic genetic circuits have been built to implement computational operations[4–10], diverse dynamic behaviors[11,12], and even coordinate multicellular actions[13–15].

The task of reprograming living cells is simplified by employing genetically encoded devices that use common input and output signals[1,2,7,8]. This allows the output of one device to be connected to the input of another to create circuits implementing more complex functionalities. Signals can take many forms, but one of the most commonly used is RNA polymerase (RNAP) flux which can be guided by promoters to specific points in a circuit's DNA[7,16]. The response function of a genetic device captures how input signals map to output signals at steady state[1,7,16]. By ensuring the response functions of two devices are compatible, that is, the range of the outputs from the first device spans the necessary range of inputs for the second device, larger circuits with desired functions can be constructed[17]. Matching of components is vital in circuits where devices exhibit switching behaviors (e.g., Boolean logic) to ensure input signals are sufficiently separated to trigger required transitions between "on" and "off" states as signals propagate through the circuit.

Although the use of characterized genetic devices has enabled the automated design of large circuits[7,18], they are often sensitive to many factors. Differences in host physiology[19–21] and interactions between genetic parts and the host cell[22–26] can all affect the response function of a device and, subsequently, its compatibility within a circuit. This makes the creation of robust genetic circuits a challenge. Even when considering controlled lab conditions, a genetic circuit often needs to be reassembled from scratch multiple times until a working combination of parts is found. This is time-consuming and costly, and often has to be repeated if the circuit is deployed into slightly different conditions or host strains.

In this work, we tackle this problem by developing genetic devices whose response functions can be dynamically tuned after circuit assembly to correct for unwanted changes in their behavior. The ability to tune/modify the steady-state input–output relationship is achieved by employing a simple regulatory motif. We show how this motif can be connected to small molecule sensors to characterize its function, and then illustrate its use in practice by integrating it into genetic NOT and NOR logic gates[27] to tune their transition points between "on" and "off" states. These capabilities make the devices more broadly compatible with other components[1,7,17], but their use comes at a cost. As we tune each device, a decrease in the dynamic range is observed and it becomes more difficult to differentiate cellular states due to variability in gene expression across a population. Mathematical modeling is used to help us better understand these limitations and derive principles to further optimize device designs. This work is a step towards the synthesis of adaptive genetic circuitry where individual components tune their response function to ensure robust system-level behaviors are maintained no matter the genetic, cellular, or environmental context.

## Results

**Constructing a tunable expression system.** To allow for the response of a genetic device to be dynamically modified, we developed a tunable expression system (TES) based on a simple regulatory motif where two separate promoters control the transcription and translation rates of a gene of interest (Fig. 1a). By using promoters as control inputs, it is possible to easily connect a TES to existing genetic components/circuitry or even endogenous

transcriptional signals within a cell. The TES contains a toehold switch (THS) that enables the translation initiation rate of the gene of interest to be varied by the relative concentration of a tuner small RNA (sRNA)[6,28]. The main component of the THS is a 92 bp DNA sequence that encodes a structural region and a ribosome binding site (RBS) used to drive translation of a downstream protein coding region. This is expressed from a promoter that acts as the main input to the TES (Fig. 1a). When transcribed, the structural region of the THS mRNA folds to form a hairpin loop secondary structure that hampers ribosome accessibility to the RBS and reduces its translation initiation rate. This structure is disrupted by a second component, a 65 nt tuner sRNA that is complementary to the first 30 nt of the THS[28]. The tuner sRNA is expressed from a second promoter, which acts as a tuner input to the device (Fig. 1a). Complementarity between the tuner sRNA and a short unstructured region of the THS enables initial binding, making it thermodynamically favorable for the sRNA to unfold the secondary structure of the THS through a branch migration process. This makes the RBS more accessible to ribosomes, which increases the translation initiation rate. Relative concentrations of the THS mRNA and tuner sRNA (controlled by the input and tuner promoters) enable the rate of translation initiation to be varied over a 100-fold range for the THS design (variant 20) we selected[28] ("Methods"). However, THS designs exist which allow for up to a 400-fold change in translation initiation rates[6,28]. We selected as main and tuner inputs for the TES the output promoters of two sensors, $P_{tet}$ and $P_{tac}$, that respond to anhydrotetracycline (aTc) and isopropyl β-D-1-thiogalactopyranoside (IPTG), respectively (Fig. 1b). Yellow fluorescent protein (YFP) was used as the output (Fig. 1b) to allow us to measure the rate of protein production in single cells using flow cytometry.

Characterization of the device was performed in *Escherichia coli* cells grown in different concentrations of aTc (input) and IPTG (tuner). Steady-state fluorescence measurements were taken using flow cytometry and promoter activities of the main and tuner inputs were measured in relative promoter units (RPUs) to allow for direct comparisons ("Methods"; Supplementary Fig. 1). A further advantage of characterizing our devices in RPUs is that the data become compatible with genetic design automation software such as Cello[7], allowing our parts to be interfaced with a large library of existing sensors and logic gates[27,29].

For a fixed tuner promoter activity, we observed a sigmoidal increase in output YFP fluorescence as the input promoter activity increased from 0.002 to 6.6 RPU (Fig. 1c). As the activity of the tuner promoter increased from 0.002 to 2.6 RPU, the entire response function shifted upwards to higher YFP fluorescence. Notably, this shift was not uniform, with larger relative increases seen for lower input promoter activities; 28-fold versus 4.5-fold for inputs of 0.002 and 6.6 RPU, respectively (Fig. 1c). Closer analysis of the flow cytometry data (Fig. 1d) showed that these changes arose from the distributions of YFP fluorescence for low and high inputs shifting uniformly together as the tuner promoter activity was increased. Therefore, even though a similar relative difference between outputs for low and high inputs (also referred to as the dynamic range) was observed for all tuner inputs, when the tuner input was low, the distributions were virtually identical to the autofluorescence of the cells (Fig. 1d). This led to even small absolute differences in the median values between low and high input states resulting in high fold changes.

Flow cytometry data also showed a significant overlap in the output YFP fluorescence distributions for low and high input promoter activities (Fig. 1d). Many applications require that on and off states in a system are well separated so that they can be accurately distinguished. To assess this feature, we calculated the fractional overlap between the output YFP fluorescence distributions for low and high input promoter activities ("Methods"). We

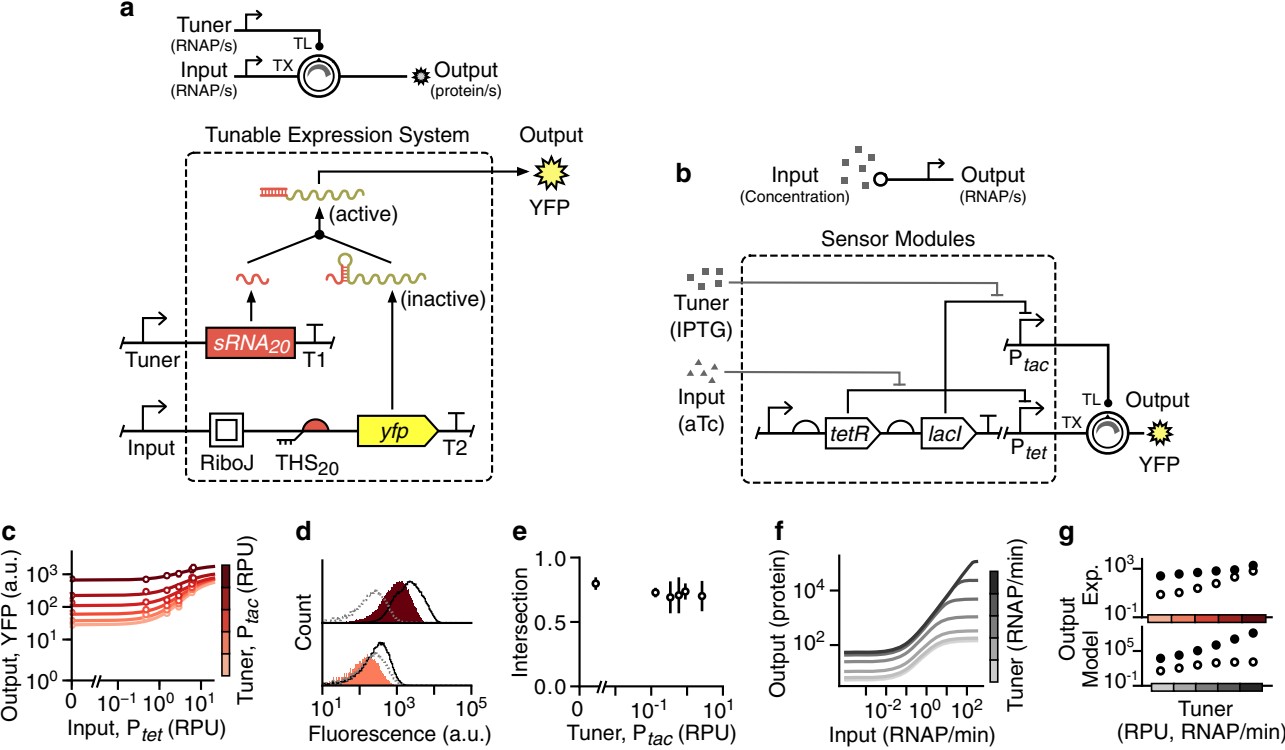

**Fig. 1 Design and characterization of a tunable expression system (TES). a** Schematic of the TES (top) and genetic implementation using a THS (variant 20)[28] to regulate translation initiation rate of an output protein (bottom, dashed box). Yellow fluorescent protein (YFP) is used as the output and T1 and T2 correspond to the transcriptional terminators L3S3P11 and L3S2P21, respectively[65]. **b** Genetic design of the sensor modules used to drive the main and tuner inputs to the TES. **c** Experimentally measured response functions for the TES. Points denote the average of three biological replicates and error bars show ±1 standard deviation. Each line shows a fitted Hill function for a fixed tuner input (color scale light–dark: 0.002, 0.03, 0.15, 0.43, 0.9, 2.6 RPU). **d** Flow cytometry distributions of output YFP fluorescence when the tuner promoter activity is low (bottom; 0.002 RPU) and high (top; 2.6 RPU). Black outlined distributions correspond to a high input promoter activity (6.6 RPU) and the filled red distributions to a low input promoter activity (0.002 RPU). Cell autofluorescence is shown by the dashed gray line. **e** Fraction of intersection between YFP fluorescence distributions for low (0.002 RPU) and high (6.6 RPU) inputs across varying tuner promoter activities. Points denote the average of three biological replicates and error bars show ±1 standard deviation. **f** Response functions from a deterministic model of the TES (Supplementary Note 1). Output shown as the steady-state protein level. Line color corresponds to the promoter activity of the tuner input (light–dark: 0.0001, 0.06, 0.3, 1.5, 7.6, 38, 190 RNAP/min). **g** Comparison of the output for high (filled circles; 6.6 RPU) and low (unfilled circles; 0.002 RPU) inputs across a range of tuner promoter activities (Experiment: 0.002, 0.03, 0.15, 0.43, 0.9, 2.6 RPU; Model: 0.0001, 0.3, 1.5, 7.6, 38, 190 RNAP/min). Source data are available in the Source Data file.

found a constant intersection of ~70% across all tuner promoter activity levels (Fig. 1e), which resulted from the similar shifts we saw in output across all input promoter activities (Fig. 1d).

To better understand these effects, we derived a deterministic ordinary differential equation (ODE) model of the system (Supplementary Note 1). Simulations using biologically realistic parameters (Supplementary Table 1) showed similar qualitative behavior to the experiments; increasing tuner promoter activity shifted the response curve to higher output protein production rates (Fig. 1f). However, unlike the experiments, increasing the tuner promoter activity resulted in a smaller increase in the fold change in the output between low and high inputs (Fig. 1g, bottom). The limiting effect that the tuner sRNA can have is a possible mechanism that could account for the nonlinear response observed in the experiments, where "on" states did not increase as quickly as "off" states as the tuner activity increased (Fig. 1g, top). Tuner sRNA concentration was fixed for each response function. Therefore, it could have been higher than the concentration of THS transcript (i.e., nonlimiting) when the main input was low, while limiting the output when the main input was high.

Another potential cause of this nonlinear response could be retroactivity that occurs when the behavior of components in a biological circuit changes once they become interconnected[30,31].

Such effects break modularity and make it difficult to predict circuit behavior. To explore this possibility, we coupled our existing model to another that is able to capture retroactivity-like effects due to shifts in ribosome allocation between endogenous genes and synthetic constructs, such as the TES (Supplementary Note 2)[23,30,31]. Ribosomes are a key cellular resource and fluctuations in their availability due to the burden of a synthetic construct can cause drops in protein synthesis rates across the cell, affecting upstream components in a circuit[20,22,24,26]. Comparisons between the original and coupled models, showed that retroactivity could have an impact for biologically realistic parameters, but only when the output caused significant cellular burden and only for the most highly expressed outputs (Supplementary Fig. 2).

**Design and assembly of a tunable genetic NOT gate.** Some genetic devices rely on the expression of proteins such as transcription factors to implement basic logic that can be composed to carry out more complex decision-making tasks[4,7,32]. One such device is a NOT gate, which has a single input and output[27]. NOT gates invert their input such that the output is on if the input is off and vice versa. Such a behavior can be implemented by using promoters as the input and output, with the input promoter driving expression of a repressor protein that binds to the DNA

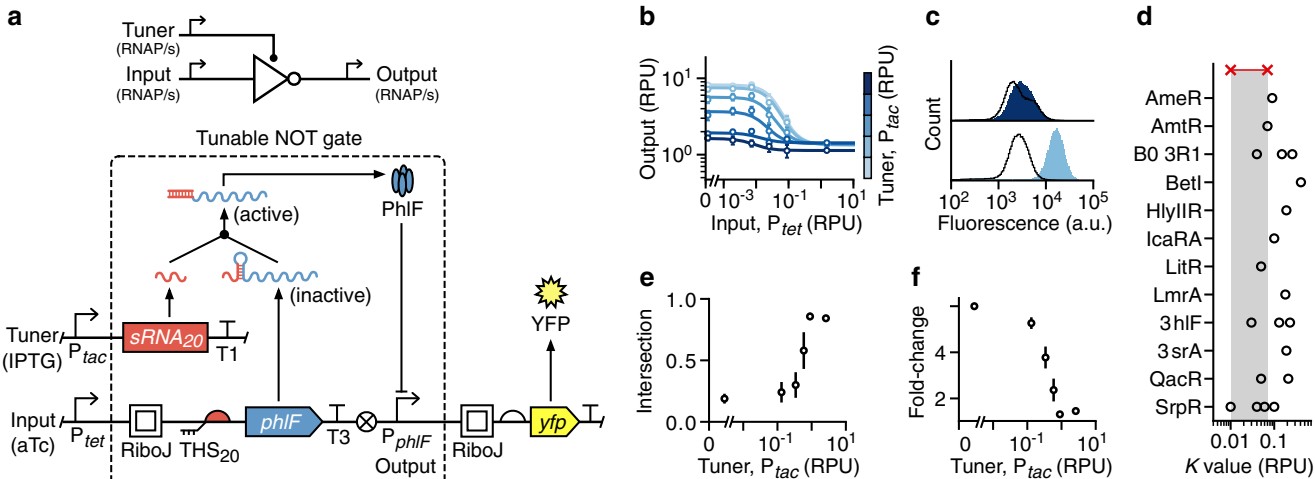

**Fig. 2 Design and characterization of a tunable NOT gate. a** Schematic of the tunable NOT gate (top) and genetic implementation embedding the TES (bottom, dashed box). Yellow fluorescent protein (YFP) expression is driven by the output promoter and T1 and T3 correspond to the transcriptional terminators L3S3P11 and ECK120033737, respectively[65]. **b** Experimentally measured response functions of the tunable NOT gate. Points denote the average of three biological replicates and error bars show ±1 standard deviation. Each line shows a fitted Hill function for a fixed tuner input (color scale light–dark: 0.002, 0.03, 0.15, 0.43, 0.9, 2.6 RPU). **c** Flow cytometry distributions of the output YFP fluorescence from the tunable NOT gate when the tuner promoter activity is low (bottom; 0.002 RPU) and high (top; 2.6 RPU). Black outlined distributions correspond to a high input promoter activity (1.5 RPU) and the filled blue distributions to a low input promoter activity (0.002 RPU). **d** Comparison of the switching point (K value) for each repressor-based NOT gate from Cello[7] (black circles) to the range achievable by the tunable NOT gate (red crosses and shaded regions). **e** Fraction of intersection between output YFP fluorescence distributions for low (0.002 RPU) and high (1.5 RPU) inputs across varying tuner promoter activities. Points denote the average of three biological replicates and error bars show ±1 standard deviation. **f** Fold change in the median output YFP fluorescence between low (0.002 RPU) and high (1.5 RPU) inputs across varying tuner promoter activities. Points denote the average of three biological replicates and error bars show ±1 standard deviation. Source data are available in the Source Data file.

of a constitutive output promoter. When the input promoter is inactive, the repressor is not synthesized and the constitutive output promoter is in an active/on state. However, once the input promoter is activated, the repressor is expressed which binds the output promoter and represses/turns off its activity.

Because the activity range of promoters varies, the transition point (where sufficient concentrations of repressor are present to cause strong repression of the output promoter) may make it impossible to connect other devices and ensure a signal is correctly propagated. For example, the output promoter of a weak sensor system acting as input to a NOT gate with a high transition point may produce insufficient repressor, causing the output promoter to be continually active. These incompatibilities can sometimes be corrected by modifying other regulatory elements in the design. In the case of a repressor-based NOT gate, while the promoters cannot be easily changed, the translation initiation rate can be varied in bacteria by altering the RBS for the repressor gene. Increasing the RBS strength causes more repressor protein to be produced for the same input promoter activity, shifting the transition point to a lower value[7,27]. While such modifications can fix issues with device compatibility, they require reassembly of the entire genetic device.

Given that the TES allows for the rates of both transcription and translation to be dynamically controlled, we attempted to create a tunable NOT gate integrating the TES to allow its response function, and crucially its transition point, to be altered after physical assembly. We chose an existing NOT gate design[27] that uses the PhlF repressor to control the activity of an output P$_{phlF}$ promoter (Fig. 2a). Expression of PhlF was controlled by the TES, replacing the YFP reporter protein in the original TES design (Fig. 1a). Unlike the TES, the tunable NOT gate uses promoters for both inputs and outputs allowing it to be easily connected to other devices that use RNAP flux as an input/output signal[7,16] (Fig. 2a).

To enable characterization of the tunable NOT gate, the output promoter P$_{phlF}$ was used to drive expression of YFP. Measurements were taken using flow cytometry for cells harboring the device in varying concentrations of aTc and IPTG, and steady-state response functions were generated (Fig. 2b, c). As expected, these showed a negative sigmoidal shape with transition points (corresponding to K values in the Hill function fitting to the experimental data) that varied over a sevenfold range (Fig. 2b). We also found that increases in the tuner promoter activity led to transitions at lower activity levels for the input promoter. The range of transition points achieved by our device covered a high proportion (35%) of the largest collection of repressor-based NOT gates built to date (total of 20 variants; Fig. 2d)[7].

These results demonstrate the ability for the TES to dynamically alter a key characteristic of a NOT gate's response function and improve its compatibility with other genetic devices. However, tuning came at a cost; it resulted in a drop in the fold change between low and high outputs (Fig. 2e) and an increase in the overlap between output YFP fluorescence distributions, making on and off states difficult to distinguish (Fig. 2f).

**Boosting sRNA levels improves device performance.** For the THS to function correctly, it is essential that the sRNA reaches a sufficiently high concentration relative to the THS transcript to ensure the associated RBS is in a predominantly exposed state[28]. In our design, the tuner promoter P$_{tac}$ has less than half the maximum strength of the main input promoter P$_{tet}$ (Supplementary Fig. 1). Furthermore, although the tuner sRNA contains a hairpin to improve its stability, sRNAs are generally more quickly degraded than normal transcripts[33,34].

To better understand the role that the THS transcript to tuner sRNA ratio had on the performance of the TES, we used our mathematical model of the system (Supplementary Note 1) to explore how various key parameters (e.g., transcription rates and

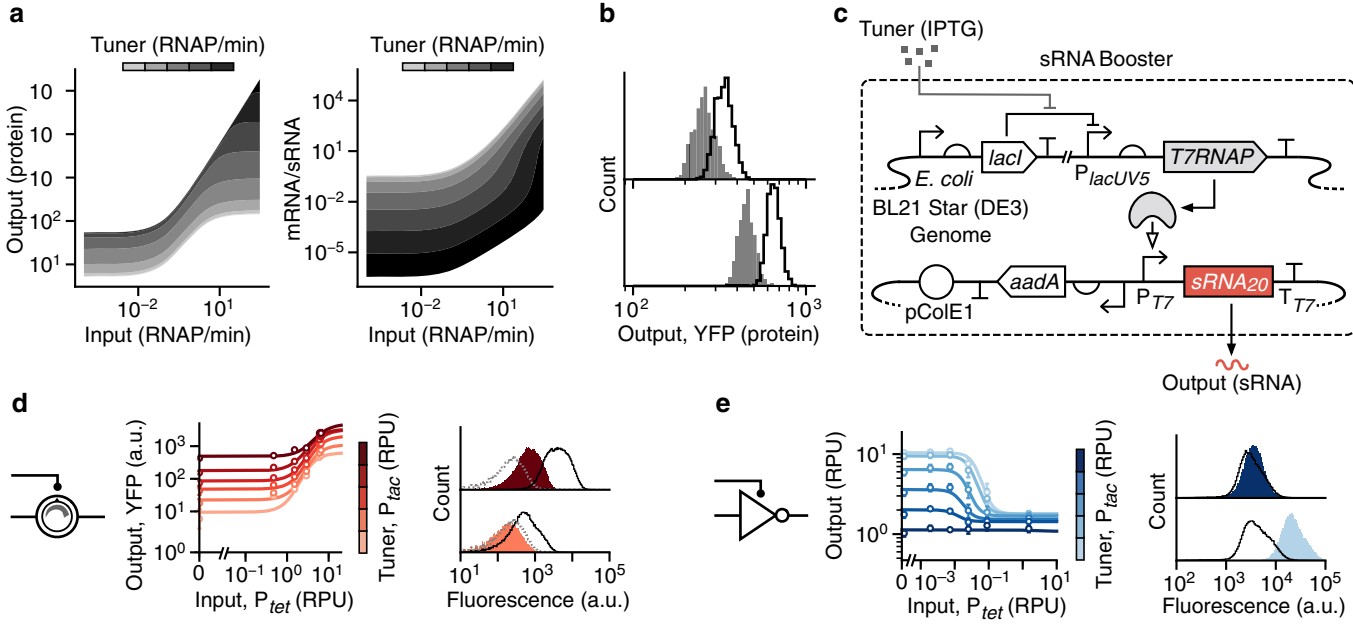

**Fig. 3 Increasing tuner sRNA transcription rate to improve device performance. a** Results of deterministic simulations of the TES model (Supplementary Note 1) showing steady-state protein output and THS mRNA to tuner sRNA ratio for a range of input and tuner promoter activities. Tuner promoter activities are shown in bands between (light–dark) 0.0001, 0.0005, 0.0024, 0.012, 0.056, 0.27, 1.3, 6.4, 31, 150, and 730 RNAP/min, respectively. **b** Stochastic simulation of the TES model (n = 4000) for low (1 RNAP/min; gray) and high (1.5 RNAP/min; green) input promoter activity. Top and bottom panels correspond to low (1.5 RNAP/min) and high (5 RNAP/min) tuner promoter activities, respectively. **c** Genetic design of the sRNA booster. The *T7RNAP* gene is encoded in the host genome and an additional plasmid contains a tuner sRNA expression unit. **d** Experimentally measured response functions (left) and flow cytometry distributions of the YFP fluorescence output (right) for the TES with the sRNA booster present. **e** Experimentally measured response functions (left) and flow cytometry distributions of the YFP fluorescence output (right) for the tunable NOT gate with the sRNA booster present. Points in all response functions denote the average of three biological replicates and error bars show ±1 standard deviation. Each line shows a fitted Hill function for a fixed tuner input (color scale light–dark: 0.002, 0.03, 0.15, 0.43, 0.9, 2.6 RPU). All flow cytometry distributions are shown for low (bottom; 0.002 RPU) and high (top; 2.6 RPU) tuner promoter activity. Black outlined distributions correspond to a high input promoter activity (6.6 RPU for the TES and 1.5 RPU for the NOT gate) and filled red/blue distributions to a low input promoter activity (0.002 RPU). Cell autofluorescence is shown by the dashed gray line. Source data are available in the Source Data file.

binding affinities) affected the response function of the device. Using biologically realistic ranges of parameters (Supplementary Table 1), we found that for lower sRNA transcription rates the output response function could be shifted to maintain a similar fold change between low and high output states (Fig. 3a). At these low THS/sRNA ratios the translation rate from the THS transcript is limited by the sRNA concentration. However, as the sRNA transcription rate increased, a transition point was seen where for low THS transcription rates the sRNA is in excess the resulting THS transcript concentration limits the output protein production rate (Fig. 3a). In contrast, at high THS transcription rates, the sRNAs become limiting again but allow for a relatively higher output protein production rate causing a larger fold change in the response function (Fig. 3a). Further stochastic modeling of the system showed that increasing sRNA transcription rate also reduced variability in the distribution of protein production rates across a population and lowered the fractional intersection between low (off) and high (on) output states (Fig. 3b).

To experimentally verify the benefit of increasing the sRNA transcription rate, we built a complementary sRNA booster plasmid that contained a high-copy pColE1 origin of replication (50–70 copies per cell)[35] and expressed the tuner sRNA from a strong viral $P_{T7}$ promoter (Fig. 3c)[36]. Transcription from $P_{T7}$ requires T7 RNA polymerase (T7RNAP). This is provided by our host strain *E. coli* BL21 Star (DE3), which has the *T7RNAP* gene under the control of an IPTG inducible $P_{lacUV5}$ promoter within its genome (Fig. 3c)[37]. Using IPTG, induction of the tuner $P_{tac}$ promoter in our devices leads to simultaneous expression of

T7RNAP from the host genome and transcription of additional tuner sRNA from the booster plasmid (Fig. 3c). As the tunable devices are encoded on a plasmid with a p15A origin of replication (~15 copies per cell; Supplementary Fig. 3)[38], we would expect that a five times higher tuner sRNA concentration is reached when the sRNA booster is present.

Cells were co-transformed with each tunable genetic device and sRNA booster plasmid, and their response functions measured (Fig. 3d, e). As predicted by the modeling, the TES performance improved with more than a doubling in the fold change across all tuner promoter activities and a >40% drop in the intersection between low and high output YFP fluorescence distributions (Table 1). For the tunable NOT gate only minor differences in performance were seen with mostly small decreases in fold change for high tuner promoter activities.

**Self-cleaving ribozymes impact toehold switch function.** In our initial designs, a RiboJ self-cleaving ribozyme was included in the TES and NOT gate to insulate the translation of the *yfp* or *phlF* genes, respectively, from different 5′ untranslated region (UTR) sequences that might be generated when using different promoters as an input (Figs. 1a and 2a)[39]. Any variable UTR sequences would be cleaved at the RiboJ site to produce a standardized mRNA with more consistent degradation and translation rates. Unfortunately, because RiboJ contains a number of strong secondary RNA structures[39,40], it is possible that the 23 nt hairpin at the 3′-end impacts the ability for the sRNA to interact with the THS, reducing the hybridization rate (Fig. 4a).

**Table 1 Performance summary of TES and tunable NOT gate.**

| Device | Design | Dynamic range (a.u.) | | Fold change | | Intersection | | K range (RPU) |
|---|---|---|---|---|---|---|---|---|
| | | Low | High | Low | High | Low | High | |
| TES | Original | 333 ± 53 | 877 ± 695 | 14 ± 1.7 | 2.4 ± 1.2 | 0.78 ± 0.06 | 0.69 ± 0.16 | – |
| | sRNA booster | 538 ± 51 | 2064 ± 1070 | 227 ± 297 | 5.7 ± 1.8 | 0.46 ± 0.04 | 0.35 ± 0.15 | – |
| | Noninsulated | 882 ± 134 | 2149 ± 409 | 445 ± 412 | 31 ± 16 | 0.26 ± 0.07 | 0.27 ± 0.06 | – |
| | Combined | 1550 ± 209 | 1712 ± 584 | 1236 ± 613 | 66 ± 54 | 0.15 ± 0.04 | 0.22 ± 0.04 | – |
| NOT gate | Original | 17280 ± 1273 | 3512 ± 286 | 6.0 ± 0.1 | 1.5 ± 0.1 | 0.19 ± 0.03 | 0.84 ± 0.02 | 0.01–0.07 |
| | sRNA booster | 22040 ± 1601 | 2170 ± 654 | 5.8 ± 0.3 | 0.9 ± 0.3 | 0.13 ± 0.07 | 0.85 ± 0.02 | 0.01–0.06 |
| | Noninsulated | 17466 ± 1926 | 4061 ± 827 | 6.8 ± 0.3 | 2.6 ± 0.4 | 0.11 ± 0.03 | 0.56 ± 0.08 | 0.02–0.04 |
| | Combined | 27751 ± 3104 | 2383 ± 165 | 6.0 ± 0.6 | 0.9 ± 0.1 | 0.08 ± 0.05 | 0.90 ± 0.03 | 0.003–0.02 |

Average values are shown ±1 standard deviation calculated from flow cytometry data for three biological replicates. The low and high columns correspond to experiments when the tuner promoter activity is 0.002 RPU and 2.61 RPU, respectively. Dynamic range is calculated as the absolute difference in YFP fluorescence between on and off inputs states. The on and off input states correspond to input promoter activities of 6.6 RPU and 0.002 RPU for the TES, and 1.5 RPU and 0.002 RPU for the NOT gate, respectively. Fold change is calculated for YFP fluorescence between on and off input states. Intersection is calculated as the fractional overlap between distributions for on and off input states. The K range gives the span of K values from Hill functions fitted to experimental data. The designs are as follows: original designs are the initial constructs (Figs. 1a and 2a), sRNA booster designs include the additional sRNA booster plasmid (Fig. 3c), the noninsulated designs have the RiboJ element removed (Fig. 4), and the combined designs have both RiboJ removed and the sRNA booster plasmid present.

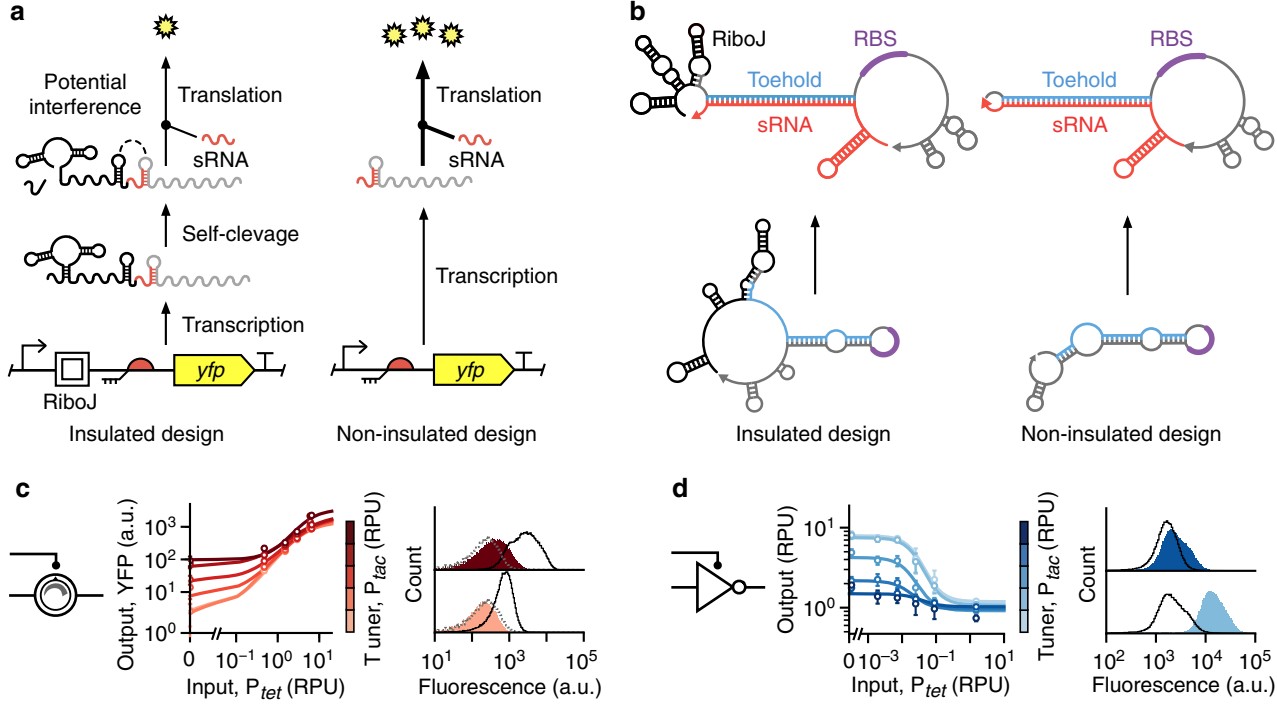

**Fig. 4 Self-cleaving ribozyme insulators affect tunable device performance. a** Original designs of both the TES and tunable NOT gate include a RiboJ insulating element, which can potentially interfere with binding of the tuner sRNA to the THS. **b** RNA secondary structure predictions for THS mRNA alone and with a complimentary tuner sRNA bound. Separate structures shown when the RiboJ insulating element is present (left) and absent (right). **c** Experimentally measured response functions (left) and flow cytometry distributions of the output YFP fluorescence (right) for the TES with the RiboJ insulator removed. **d** Experimentally measured response functions (left) and flow cytometry distributions of the YFP fluorescence output (right) for the tunable NOT gate with the RiboJ insulator removed. Points in all response functions denote the average of three biological replicates and error bars show ±1 standard deviation. Each line shows a fitted Hill function for a fixed tuner input (color scale light–dark: 0.002, 0.03, 0.15, 0.43, 0.9, 2.6 RPU). All flow cytometry distributions are shown for low (bottom; 0.002 RPU) and high (top; 2.6 RPU) tuner promoter activity. Black outlined distributions correspond to a high input promoter activity (6.6 RPU for the TES and 1.5 RPU for the NOT gate) and filled red/blue distributions to a low input promoter activity (0.002 RPU). Cell autofluorescence is shown by the dashed gray line. Source data are available in the Source Data file.

To assess whether the RiboJ insulator might affect the stability of secondary structures that are crucial to the TES's function, we performed thermodynamic modeling of the binding between the THS mRNA and the tuner sRNA for variants of the TES design with and without RiboJ present ("Methods"). Simulations predicted a 40% drop in Gibbs free energy of the reactants when RiboJ was removed (−40.5 kcal/mol with versus −65 kcal/mol without RiboJ;

Fig. 4b). This suggests that binding between sRNAs and the THS may be hampered by interactions with the RiboJ insulator, lowering the effective translation initiation rate of the RBS controlled by the THS and subsequently the performance of the devices.

To experimentally test these predictions, noninsulated variants of the TES and tunable NOT gate were constructed in which RiboJ was removed. Characterization of these devices showed

major improvements in overall performance (Fig. 4c, d). The TES saw more than a doubling in the dynamic range and a tenfold increase in the fold change between "on" and "off" states across low and high tuner activity levels (Table 1). In addition, the fraction of intersection of the output YFP fluorescence distributions dropped by >50%. The tunable NOT gate saw more modest improvements with a 73% increase in the fold change at high tuner activity levels, but an overall drop of 66% in the range of transition points ($K$ values) that could be achieved (Table 1). These results highlight an important consideration often ignored. When using RNA-based devices that require the proper formation of secondary structures, care must be taken to ensure multiple parts do not interfere with each other, leading to cryptic failure modes.

Another counterintuitive change in the TES's response function after RiboJ removal was the large drop in output YFP fluorescence from 26 to 3 arbitrary units (a.u.) when no input or tuner was present (Fig. 4c). Similar drops of between 4- and 11-fold were also seen for higher tuner promoter activities. Given that binding of a tuner sRNA to the THS mRNA should be less hampered when RiboJ is absent, an increase rather than a decrease in output protein production would be expected. A possible explanation is that the stability of the THS mRNA decreased after RiboJ was removed. This is supported by recent results that have shown the RiboJ insulator both stabilizes its mRNA and also boosts the translation initiation rate of a nearby downstream RBS[41]. The precise mechanisms for this are not well understood but it is thought that the structural aspect of the RiboJ at the 5′-end of an mRNA inhibits degradation by exonucleases, whilst the hairpin at the 3′-end exposes the nearby RBS by reducing the chance of unwanted secondary structure formation[39,40].

Finally, we combined the noninsulated designs with the sRNA booster plasmid to see whether further improvements could be made (Table 1). For the TES, we found that the dynamic range had plateaued, with only moderate increases at low tuner promoter activities. In contrast, the fold change between low and high outputs more than doubled across tuner promoter activities when compared to the noninsulated design, and a further drop of >18% was seen in the fractional intersection between the YFP fluorescence distributions for these output states. The tunable NOT gate showed minor decreases in performance for many of the measures (Table 1). However, the inclusion of the sRNA booster likely increased overall PhlF concentrations as the transition points from an "on" to an "off" state shifted far below what had been seen for all other designs. This would make this specific design of value for uses where a weak input signal needs to be inverted and amplified simultaneously.

**Towards complex tunable logic**. To create larger genetic circuits that implement complex logic, it is vital that a sufficiently diverse set of logic gates are available for use. Because a NOT gate alone has limited capabilities, we further modified its design to create a tunable two-input NOR gate[7,27]. The output of a NOR gate is on only when both inputs are off (Fig. 5b) and crucially this type of gate is functionally complete (i.e., any combinatorial logic function can be implemented using NOR gates alone). In our new device, we added a further inducible input promoter, $P_{BAD}$, directly before the existing $P_{tet}$ input promoter, and included the associated sensor system (*araC* gene) to allow activity of the $P_{BAD}$ promoter to be controlled by the concentration of L-Arabinose (Ara) (Fig. 5a). Our modifications were made to the original NOT gate design that included the RiboJ insulator because this produced the largest tunable range for the "on" to "off" transition point.

To assess the function of the tunable NOR gate, the activities of both input promoters $P_{BAD}$ and $P_{tet}$, and the tuner promoter $P_{tac}$ were varied by culturing cells harboring the device in different concentrations of Ara, aTc and IPTG, respectively ("Methods"). The two-dimensional response functions (Fig. 5c) showed that NOR logic was successfully implemented and that the transition point from low to high output for both inputs was simultaneously shifted to lower input promoter activities when the tuner promoter was highly active (Fig. 5c, right panel). Considering each input promoter separately, the transition point between "on" and "off" states shifted by 16- and 6-fold for $P_{BAD}$ and $P_{tet}$, respectively.

Unlike the NOT gate, even at high tuner promoter activities, the dynamic range was better maintained, dropping at most 35%, and the fold change between "on" and "off" states remained above 4-fold and 8-fold for low and high tuner promoter activities, respectively (Supplementary Table 3). Furthermore, the improved separation of these states led to smaller intersections in the output YFP distributions compared to the NOT gate. This was especially evident when comparing NOR gate states where both input promoters were simultaneously on or off with only a ~5% intersection observed (Supplementary Table 3).

The cause of this improvement is not clear but may relate to the $P_{BAD}$ promoter insulating expression of the *phlF* gene from transcriptional read-through originating from the tuner sRNA transcriptional unit that is located directly upstream in the DNA (Supplementary Fig. 3). Without this insulating effect, read-through would cause elevated expression of PhlF, even when the input promoters are off, and potentially lead to a partial switch in the output when the tuner promoter is active (as seen for the original NOT gate, Fig. 2b). Such a mechanism could also account for the elevated output levels for the TES when the input promoter was off and the tuner promoter activity increased (Fig. 1c).

## Discussion

In this work, we developed a class of genetic device whose response function can be dynamically tuned. This was achieved by constructing a TES to separately control the transcription and translation rate of a gene. We demonstrate how the TES can be used to shift the "on" and "off" output states of a sensor by 4.5- and 28-fold, respectively (Fig. 1) and incorporated into NOT and NOR gates to alter their transition point between on and off output states over a greater than 6-fold range (Fig. 2). Unfortunately, the performance of the tunable devices varied for differing tuner inputs, leading to a trade-off between performance and the level of tuning required. Mathematical modeling revealed (1) the importance of ensuring sufficient tuner sRNA is present to fully activate the THS (Fig. 3) and (2) the presence of possible detrimental interactions between a self-cleaving ribosome and the THS (Fig. 4). Modified designs addressing these concerns showed improved performance for the TES, but only minor improvements in the fold change of the tunable NOT gate when the self-cleaving ribozyme was removed (Table 1). In contrast, the NOR gate behaved more consistently across tuner activity levels and displayed better separation between "on" and "off" states. To the best of our knowledge the simultaneous control of transcription and translation to tune the response function of a genetic device is original, making this work a valuable resource for others to build on. Furthermore, unlike other attempts at tuning the response of devices through mutation of protein components to alter catalytic rates[42], our method allows for dynamic changes to a response function using simple to control transcriptional signals.

A difficulty when using THSs to regulate gene expression is that high concentrations of sRNA are required to achieve a strong

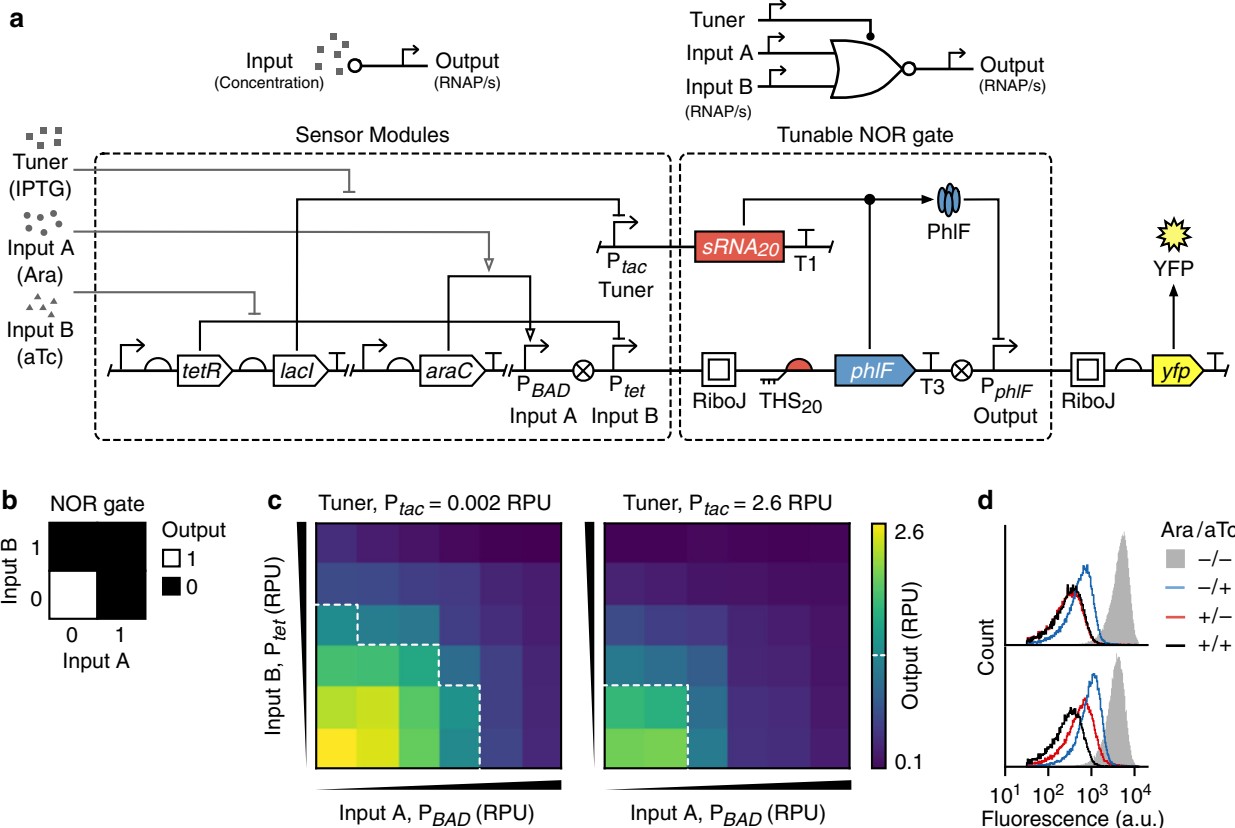

**Fig. 5 Design and characterization of a tunable NOR gate. a** Schematic of all the sensor systems used (top, left), the tunable NOR gate (top, right), and their genetic implementation (bottom, dashed boxes). Yellow fluorescent protein (YFP) expression is driven by the output promoter and T1 and T3 correspond to the transcriptional terminators L3S3P11 and ECK120033737, respectively[65]. **b** Function of a two-input NOR gate. **c** Heatmaps showing the output of the tunable NOR gate for varying input promoter activities (Input A–$P_{BAD}$: 0.008, 0.003, 0.15, 0.5, 2.5, 3.1 RPU; Input B–$P_{tet}$: 0.05, 0.5, 1.6, 3.1, 6.4, 7.5 RPU) and for low (left) and high (right) tuner promoter activities. Output promoter activities shown are average values calculated from flow cytometry data for three biological replicates. White dashed line shows an output of 1.2 RPU and denotes the transition point from a high to low output. **d** Flow cytometry distributions of the output YFP fluorescence for tuner promoter activities of 0.002 RPU (bottom) and 2.6 RPU (top). The four distributions correspond to combinations of the absence and presence of L-Arabinose (10 mM) and aTc (50 ng/mL). Source data are available in the Source Data file.

enough activation of mRNA translation. This stems from the regulatory mechanism which relies on base-pairing of the sRNA to THS, placing limits on the binding affinity that can be achieved. A possible means of increasing the affinity between these species would be to exploit RNA chaperones such as Hfq[43,44]. In prokaryotes, sRNAs that associate with Hfq play a variety of roles from inhibiting and activating translation, to affecting the stability of a target mRNA[45–47]. In some cases, these effects are significant; it has been shown in vitro that Hfq increases by 30- to 50-fold the binding affinity of the DsrA sRNA to the leader of the *rpoS* mRNA[48]. Designing de novo sRNAs that bind to Hfq to increase their affinity to a target mRNA has been shown for both activation[43] and inhibition[44] of translation initiation. In both cases, Hfq binds a scaffold from an endogenous gene (e.g., *micC*) which is fused with a targeting sequence (e.g., that found on the sRNA). This approach could be employed in future TES designs. In fact, previous work that used Hfq associated sRNAs to implement a metabolically cheap negative feedback control loop created a useful repressive tuning element that could be directly used in our system[44]. By combining the findings from that study with ours and incorporating recent improvements in THS design[6], it should be possible to make further strides towards high-performance tunable genetic devices.

An interesting future direction opened up by the adaptive nature of our devices is the possibility to incorporate many of

them into larger circuits. This would allow multiple parts of a circuit to be tuned simultaneously to maximize component compatibility and optimize system behavior. Unlike a typical design–build–test cycle that requires the reassembly of a genetic circuit if malfunctions are detected, this work supports a design–build–test–tune cycle where costly reassembly can be avoided. Rather than reassembling a circuit, parts can instead be dynamically tuned until they work correctly in unison. In this context, applying sensitivity analysis during circuit design would allow us to identify specific components where even small deviations in behavior would adversely impact the overall circuit function[49]. These would be ideal candidates to be encoded using tunable devices to allow for tweaking at these critical points.

The additional tuner inputs in our devices raise some practical challenges. Systems composed of numerous tunable devices will require a large number of tuner inputs to be controlled simultaneously. If external signals are to be used then a unique sensor is required for each tuner input, as well as the capability to be able to control the environment to provide the correct set of input signals over time. Although the range of small molecule[29] and light based[42,50] sensing systems has grown over recent years, the ability to control many environmental factors (e.g., small molecule concentrations) simultaneously remains difficult. However, external control is not the only way to tune the behavior of these devices. The use of promoters as inputs allows them to be

controlled by connecting them directly to the many transcriptional signals used natively in a cell. This offers the advantage of tapping into the cells innate capacity to sense and respond to its environment. Alternatively, if an adaptive circuit is not required, sensors controlling the tuning inputs could be replaced once a working configuration is found with constitutive promoters of an identical strength. This would reduce the reassembly required to a single step once the correct combination of tuning inputs is found.

When designing our tunable devices, we observed deviations between the experimental and modeled responses. This may be due to retroactivity[30,31], where expression of the output reporter protein places a significant burden on the host cell (Supplementary Note 2). Recently, there has been increased interest in the role of burden[51] and attempts made to mitigate its effect[52]. One approach has been to implement resource allocation schemes based on split exogenous RNAPs[53]. This limits the maximum burden a circuit can impose by providing fixed size pools of transcriptional resources that are orthogonal to the endogenous ones. Because our devices can have their responses dynamically tuned, they could be used to boost the expression of downstream components to mitigate retroactivity effects or even be used to cap the maximum levels of resource that can be used by a circuit.

For synthetic biology to have a broad impact outside of the carefully controlled conditions of a lab, it is vital that we are able to build adaptive genetic circuits that can continue to maintain their functionality when exposed to unexpected environmental changes or shifts in host cell physiology[54]. By combining advances in biological control engineering[52,54–58] with the tunable genetic devices developed in this work, bioengineers have a complementary set of tools capable of taking steps towards this goal.

## Methods

**Strains and media.** Cloning was performed using *E. coli* strain DH5-α (F⁻ *endA1 glnV44* thi-1 *recA1 relA1 gyrA96 deoR nupG purB20* φ80d*lacZ*ΔM15 Δ(*lacZYA–argF*)U169, *hsdR*17(r_K⁻m_K⁺), λ⁻) (New England Biolabs, C2987I). Device characterization was performed using BL21 Star (DE3) (F⁻ *ompT hsdS_B* (r_B⁻, m_B⁻) *gal dcm rne*-131 [DE3]) (Thermo Fisher Scientific, C601003). For cloning, cells were grown in LB Miller broth (Sigma-Aldrich, L3522). For device characterization, cells were grown in M9 minimal media supplemented with glucose containing M9 salts (6.78 g/L Na₂HPO₄, 3 g/L KH₂PO₄, 1 g/L NH₄Cl, 0.5 g/L NaCl) (Sigma-Aldrich, M6030), 0.34 g/L thiamine hydrochloride (Sigma T4625), 0.4% D-glucose (Sigma-Aldrich, G7528), 0.2% casamino acids (Acros, AC61204-5000), 2 mM MgSO₄ (Acros, 213115000), and 0.1 mM CaCl₂ (Sigma-Aldrich, C8106). Antibiotic selection was performed using 50 μg/mL kanamycin (Sigma-Aldrich, K1637) or 50 mg/mL spectinomycin (Santa Cruz Biotechnology, sc-203279). Induction of sensor systems was performed using aTc (Sigma-Aldrich, 37919), IPTG (Sigma-Aldrich, I6758), and L-Arabinose (Ara) (Sigma-Aldrich, A3256).

**Genetic device synthesis and assembly.** Plasmids containing the TES (pVB001) and tunable NOT gate devices (pVB002) were constructed by a combination of gene synthesis (GeneArt, Thermo Fisher Scientific) and PCR of existing plasmids to generate linear fragments with regions of homology between subsequent parts. Gibson assembly (New England Biolabs, E2611S) was then used to assemble these into the final plasmids. Supplementary Table 4 provides details of the synthesized DNA fragments (TES-P1 and TES-P2), and Supplementary Table 5 provides details of the primers and their templates used to generate all linear fragments for each plasmid design. Removal of RiboJ from the TES (pVB001) and NOT gate (pVB002) was achieved by PCR of the relevant design using primers F_RiboJ_Rem and R_RiboJ_Rem (Supplementary Table 5) and subsequent circularization by standard Golden Gate assembly (New England Biolabs, E1601S) to create the plasmids pVB003 and pVB004, respectively. The plasmid used to boost tuner sRNA levels (pVB005) was fully synthesized (GeneArt, Thermo Fisher Scientific). The plasmid containing the tunable NOR gate device (pVB006) was constructed by first PCR amplification of the pAN1720 plasmid (without the *lacZα* region normally used for blue/white screening) using primers containing an EcoRI restriction site at the 5′-end and an NotI restriction site at the 3′-end (F_pAN1720_EcoRI and R_pAN1720_NotI; Supplementary Table 5). The tunable NOR gate DNA insert was synthesized in three parts, NOR-P1, NOR-P2, and NOR-P3 (Integrated DNA Technologies), which were then assembled using a standard Golden Gate assembly method (New England Biolabs, E1601S) to create a full-length linear insert.

This was designed to contain complementary EcoRI and NotI restriction sites to the amplified pAN1720 fragment. Both linear DNA fragments were finally used in a standard restriction digest using EcoRI (New England Biolabs, R3101) and NotI (New England Biolabs, R3189), and then a ligation reaction (New England Biolabs, M0202S) used to assemble the complete pVB006 plasmid. All plasmids were sequence verified by Sanger sequencing (Eurofins Genomics). Annotated plasmid maps of all devices are provided in Supplementary Fig. 3 and Supplementary Data 2.

**Genetic device characterization.** Single colonies of cells transformed with the appropriate genetic constructs were inoculated in 200 μL M9 media supplemented with glucose and necessary antibiotics for selection in a 96-well microtiter plate (Thermo Fisher Scientific, 249952) and grown for 16 h in a shaking incubator (Stuart, S1505) at 37 °C and 1250 rpm. Following this, cultures were diluted 9:1600 (15 μL into 185 μL, with 15 μL of this dilution loaded into 185 μL) in glucose supplemented M9 media with necessary antibiotics for selection and grown for 3 h at the same conditions. Next, the cultures were diluted 1:45 (10 μL into 140 μL) into supplemented M9 media with necessary antibiotics for selection and any required inducers in a new 96-well microtiter plate (Thermo Fisher Scientific, 249952) and grown at 37 °C and 1250 rpm for 5 h. Finally, the cells were diluted 1:10 (10 μL into 90 μL) in phosphate-buffered saline (Gibco,18912-014) containing 2 mg/mL kanamycin to halt protein translation and incubated at room temperature for 1 h to allow for maturation of the YFP before performing flow cytometry.

**Flow cytometry.** YFP fluorescence of individual cells was measured using an Acea Biosciences NovoCyte 3000 flow cytometer equipped with a NovoSampler to allow for automated collection from 96-well microtiter plates. Data were collected using the NovoExpress software. Cells were excited using a 488 nm laser and measurements were taken using a 530 nm detector. A flow rate of 40 μL/min was used to collect at least 10⁵ cells for all measured conditions. Automated gating of events using the forward (FSC-A) and side scatter (SSC-A) channels was performed for all data using the FlowCal Python package version 1.2[59] and the density2d function with parameters: channels = ["FSC-A", "SSC-A"], bins = 1024, gate_fraction = 0.5, xscale = "logicle", yscale = "logicle", and sigma = 10.0. A demonstration of this automated approach is shown in Supplementary Fig. 4.

**Autofluorescence correction.** To measure YFP fluorescence from our constructs it was necessary to correct for the autofluorescence of cells. An autofluorescence control strain containing the pAN1201 plasmid[7], which does not express YFP but contains the same backbone as our genetic devices, was measured using flow cytometry under the same culturing conditions as for characterization. Measurements were taken from three biological replicates and an average of the medians of the gated distributions was subtracted from the gated YFP fluorescence flow cytometry data of the characterized devices, as in previous work[7].

**Characterization of sensor systems.** To allow for inputs to our devices to be controlled in standardized RPUs[7,60], calibration curves for the two sensor systems were generated to enable a conversion between a chemical inducer concentration and the relative promoter activity of each sensors' output promoter (P_tac and P_tet). Cells transformed with plasmids pAN1718 and pAN1719 for P_tac and P_tet, respectively, and the pAN1717 RPU standard[7], were cultured in the same way as the characterization experiments. Flow cytometry was used to measure YFP fluorescence which was further corrected for cell autofluorescence. RPU values were then calculated by dividing the YFP output from the sensor by the YFP output from the RPU standard and a Hill function fitted to the resultant data (Supplementary Fig. 1).

**Quantifying histogram intersections.** The fraction of intersection $H$ between two histograms (e.g., flow cytometry fluorescence distributions), $x$ and $y$, was calculated using,

$$H(x,y) = \sum_{i=1}^{n} \frac{\min(x_i, y_i)}{x_i}, \tag{1}$$

Here histograms $x$ and $y$ are divided into $n$ bins that correspond to identical ranges of values for each, with $x_i$ and $y_i$ denoting the value of bin $i$ for histogram $x$ and $y$, respectively.

**Predicting RNA binding and secondary structure.** To predict the binding and secondary structure of the THS and tuner sRNA (Fig. 3), thermodynamic modeling was performed using the NUPACK web application[61]. All simulations were run using the parameters: nucleic acid = RNA, temperature = 37 °C and the concentration of THS mRNA was set to $5 \times 10^{-4}$ μM. The switch sequence mRNA and the switch sequence mRNA with an upstream cleaved RiboJ were simulated independently with additional parameters strand species = 1 and a maximum complex size = 1. The THS mRNA with and without an upstream RiboJ sequence were also simulated in the presence of trigger sRNA set to a concentration of $7 \times 10^{-5}$ μM with additional parameters: strand species = 1 and a maximum complex size = 1. Full sequences are given in Supplementary Table 2.

**Computational analyses and data fitting**. All general computational analyses and plotting were performed using Python version 3.6.6, NumPy version 1.16, Pandas version 0.24, and matplotlib version 3.1. Response functions for the TES designs were generated by fitting median values of YFP fluorescence from flow cytometry data to a Hill function of the form

$$y = y_{\min} + (y_{\max} - y_{\min})\frac{x^n}{K^n + x^n}, \qquad (2)$$

where $y$ is the output YFP fluorescence (in a.u.), $y_{\min}$ and $y_{\max}$ are the minimum and maximum output YFP fluorescence (in a.u.), respectively, $K$ is the input promoter activity (in RPU units) at which the output is halfway between its minimum and maximum, $n$ is the Hill coefficient, and $x$ is the input promoter activity (in RPU units). Response functions for the tunable NOT gates were generated in a similar way using a Hill function of the form

$$y = y_{\min} + (y_{\max} - y_{\min})\frac{K^n}{K^n + x^n}. \qquad (3)$$

Fitting of data was performed using nonlinear least squares and the curve_fit function from the SciPy.integrate Python package version 1.1.

**Numerical simulation**. The deterministic ODE model (Supplementary Note 1) was simulated using the odeint function of the SciPy.integrate Python package version 1.1 with default parameters. The delay differential equations (Supplementary Note 2) were simulated using the DifferentialEquations module version 6.10 using the Bogacki–Shampine 3/2 method running in Julia version 1.3. Stochastic simulations of the biochemical model (Supplementary Note 1) were performed using the tau-leap method in COPASI version 4.24 with the following settings: number of iterations (simulations) = 4000, duration = 100 min, interval size = 1 min, number of intervals = 100, and the starting in steady-state option selected. Initial steady-state conditions for the simulation are calculated automatically by COPASI using a damped Newton method.

**Visualization of genetic designs**. All genetic diagrams are shown using Synthetic Biology Open Language Visual (SBOL Visual) notation[62]. SBOL Visual diagrams were generated using the DNAplotlib Python package[63,64] version 1.0 which were then annotated and composed with OmniGraffle version 7.9.2.

**Reporting summary**. Further information on research design is available in the Nature Research Reporting Summary linked to this article.

## Data availability

Systems Biology Markup Language (SBML) file implementing a model of the TES can be found in Supplementary Data 1. Annotated sequence files in GenBank format for all plasmids are available in Supplementary Data 2. All plasmids are available from Addgene (#127185–127189, 140327). Flow cytometry data are available at the Open Science Framework https://osf.io/dw4fp/ (DOI: https://doi.org/10.17605/OSF.IO/DW4FP). All other relevant data are available from the authors upon request. Source data for the main and supplementary figures and tables are available in the Source Data file.

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

## Acknowledgements

The authors wish to acknowledge the assistance of Dr Andrew Herman and Dr Lorena Sueiro Ballesteros of the University of Bristol Faculty of Biomedical Sciences Flow Cytometry Facility, and Bettina Frank of the Anderson Lab. This work was supported by BrisSynBio, a BBSRC/EPSRC Synthetic Biology Research Centre grant BB/L01386X/1 (M.d.B., T.E.G.), EPSRC/BBSRC Centre for Doctoral Training in Synthetic Biology grant EP/L016494/1 (V.B.), the EU Horizon 2020 research project COSY-BIO grant 766840 (M.d.B), and a Royal Society University Research Fellowship grant UF160357 (T.E.G.)

## Author contributions

T.E.G. conceived of the study. V.B., T.E.G., and G.A.M. performed the experiments. V.B., M.d.B., and T.E.G. developed the mathematical models. V.B. analyzed the data. T.E.G., V.B., M.d.B., and G.A.M. wrote the manuscript.

## Competing interests

The authors declare no competing interests.
