## [Peer Review File · Nature Communications]

Reviewers' Comments:

Reviewer #1:

Remarks to the Author:

Bartoli, di Bernardo and Gorochowski describe the development of "Tunable Genetic Devices" that use a second input to tune expression of an output. Central to their design is a positively regulating sRNA (expressed in the presence of a "tuner" input) which enables translation of a transcript (expressed under the presence of another input).

Their first result shows that the operation is as expected, but that when the tuner is full-on, there is significant overlap between low and high input. This is improved in the end, by increasing the available sRNA using a strong promoter.

The paper also describes how this can be applied to tune the input/output response of a NOT gate; and how the inclusion or not of ribozymes - which are in general thought to be helpful for insulating transcripts - can affect the performance of this and other devices. I felt this last part was not central to the story, but useful to have.

Overall the paper is interesting and several aspects are well thought out and well designed. I have some questions.

1. The overall design of the TES requires high levels of sRNA in order to operate. Some discussion on how Hfq-dependent positive regulation would affect this would be useful, if someone decided to design them that way [10.1073/pnas.1004435107](https://doi.org/10.1073/pnas.1004435107).
2. I found the discussion around lines 105-111 confusing, but this was clarified again from line 125 onward. It would be useful to show clearly that the YFP fluorescence (log-normal) distributions "shift uniformly together" and model the effects of the limiting effect of tuner sRNA to show this comparison. Also useful would be modelling that demonstrates the "uniform shift" in the distributions.
3. I think Equation S5 in the model is incorrect; it should be $C \rightarrow C+P$, and the model equations and simulations need to be repeated to verify the same conclusions.
4. Please compare the results of the TES to the design in doi.org/10.1093/nar/gky828 where an inhibiting sRNA was used to tune the feedback strength of an autorepressor.
5. Also, please consider how retroactivity could be part of the reason that a difference in fold-change between low and high outputs (lines 178-180) was observed.
6. The sentences on lines 25 and 110 don't read well; there are also some typos, such as "turner" on line 115, "P_tac" on line 185 etc. Please improve the Figure legends and consider increasing the size of some plots in the figures - for example, the "left and right distributions" in 1D refer to the top, and are indistinguishable in color.

Reviewer #2:

Remarks to the Author:

Review of "Tunable genetic devices through simultaneous control of transcription and translation" by Vittorio Bartoli, Mario di Bernardo, and Thomas Gorochowski.

This paper proposes a method to tune the (steady state) expression level of a protein, let us say p , as a function of an (inducer) input, let us say u . This tuning is performed by adjusting a parameter which is set by means of an auxiliary (inducer) input, let us say v . Thus, $p = f(u,v)$. The objective

is to use v in order to improve the range and switching quality of the function $p = F(u,v)$ as a function of u . A sharp response is particularly desirable for the effective implementation of binary logic gates.

Mathematically, the idea is very simple: u and v induce production of intermediates, and these intermediates can bind to produce a complex; it is this complex that produces the output p . The function $f(u,v)$ saturates when one of u or v is large and the other one is fixed, since there is only so much complex that can be produced by a 1:1 stoichiometry. This explains the graphs that are obtained numerically. (In fact, this explanation predicts a similar plot for p as a function of v , with the roles of input and tuner exchanged.) The results are mathematically very simple and not surprising.

This idea is implemented as follows: the input u controls transcription of an mRNA that cannot be translated unless a specific sRNA (whose transcription is controlled by v) binds to it, thus forming a complex that makes the mRNA's RBS more accessible. The translated protein (say GFP) is the output.

It would appear to me that this idea owes much to ref 6 (ribocomputing devices in paper by groups of Pam silver, Jim Collins, and Peng Yin in Nature 2017, as well as many previous papers by that group). Thus, at first sight, this paper is nice but more at the level of an iGEM project, putting parts together for illustrating a simple mathematical idea. I will defer to experts on mRNA networks to value the novelty in that regard.

The question for me is how important this work is as a general tool in synthetic biology. Much as I like the idea, I have serious doubts in that regard, because I cannot see how this will scale and/or fit in a "modular" design architecture (which the authors use as a motivation).

Issues of retroactivity and resource competition are not addressed, and that's a serious impediment to scale-up of these systems, as discussed for example in [29]. The system, by design, is based on (de-)inhibiting translation. Thus, competition for ribosomes is an especially critical issue. With more gates implemented, this will be a nontrivial issue.

Scaling up will also be extremely difficult given the need to have a "tuner" input at each node in a network. (To their credit, the authors are aware of this. Still, it makes the approach very limited.)

The heterogeneity of behavior at the single-cell level (overlap of input-low and input-high distributions) makes the system not very useful, in my opinion, in situations where the wrong "on" vs "off" state is undesirable even if this happens only in a fraction of the population. (For example, if "on" means inducing the release of a toxin, in an engineered therapeutic cell, but a large proportion of cells do so in the wrong environment.) As far as I can see, tuning does not help with this. (Perhaps a refinement of the construct can result in more homogeneity of behavior?)

The construction of a NOT gate is of some interest, but I believe that the paper would be far more interesting if some nontrivial function, say an AND gate, could be implemented and shown how to tune (with several tuners, if needed).

Another aspect that could make the paper far more interesting is showing how tuning might be used to diminish fan-out effects. For example, a higher and more homogeneous "high" state might diminish the strength of the "virtual repression" between two competing targets of a given "upstream" (and tunable) module.

Remarks:

The construct in this paper reminds me of the post-transcriptional regulation in the core fragment / sigma fragment system in Segall-Shapiro et al., Molecular Systems Biology 2014, which also

aims at tuning production through a “resource allocator” acting as a tuner (though in a different context, with many targets, and obviously with completely different biological parts, as it has to do with RNAP’s and not sRNA’s and mRNA’s). The authors might want to comment on this analogy.

Some more details on the stochastic modeling could be given, to facilitate reproducibility: how much of each sample path was dropped to assume steady state, how many runs, etc.

I do not understand how normalizing into RPU’s helps when interconnecting systems. While I understand Endy et al.’s reason for normalizing gene expression numbers by means of a comparison with a standard gene expressed in the same temperature, cell, etc., it is unclear to me how using RPU’s helps in ensuring the well-matching of interconnections. A citation or more explanation would be useful.

Typos/edits:

lines 25/26: one of the most commonly used is RNA polymerase (RNAP) flux with promoters are used to guide

-- I cannot parse this... maybe “in which promoters...”?

Mathematical typo: equation (S12) is missing a term $-kC-C$.

Response to reviewers (NCOMMS-19-24801)

Please find below our response to all of the reviewers' comments. We believe that all of their concerns have been addressed. In the revised manuscript, sentences containing changes have been highlighted in yellow.

Reviewer #1:

Bartoli, di Bernardo and Goroehowski describe the development of "Tunable Genetic Devices" that use a second input to tune expression of an output. Central to their design is a positively regulating sRNA (expressed in the presence of a "tuner" input) which enables translation of a transcript (expressed under the presence of another input).

Their first result shows that the operation is as expected, but that when the tuner is full-on, there is significant overlap between low and high input. This is improved in the end, by increasing the available sRNA using a strong promoter.

The paper also describes how this can be applied to tune the input/output response of a NOT gate; and how the inclusion or not of ribozymes - which are in general thought to be helpful for insulating transcripts - can affect the performance of this and other devices. I felt this last part was not central to the story, but useful to have.

Overall the paper is interesting and several aspects are well thought out and well designed. I have some questions.

We are grateful to the Reviewer for their careful assessment of our work and are glad that they recognized the interesting and important questions we are attempting to address.

1. The overall design of the TES requires high levels of sRNA in order to operate. Some discussion on how Hfq-dependent positive regulation would affect this would be useful, if someone decided to design them that way 10.1073/pnas.1004435107.

The Reviewer raises a very interesting point that could help reduce the levels of sRNA needed by exploiting the natural function of the Hfq complex to increase the binding affinity of sRNA and target mRNA. As this could be an interesting future direction, a new paragraph has been added to the Discussion section outlining how this approach might be used to design complementary devices with improved performance.

2. I found the discussion around lines 105-111 confusing, but this was clarified again from line 125 onward. It would be useful to show clearly that the YFP fluorescence (log-normal) distributions "shift uniformly together" and model the effects of the limiting effect of tuner sRNA to show this comparison. Also useful would be modelling that demonstrates the "uniform shift" in the distributions.

We believe that the confusion of the Reviewer was in part due to our earlier structuring of these paragraphs. We apologize if this caused confusion. To fix this problem, we have reordered these paragraphs to ensure that information required for interpretation is provided as soon as necessary.

We have also added a new panel to Figure 1 that shows how the output for on and off input states from both the flow cytometry and model behave, specifically, in regard to the near uniform shifts in output. New text explaining these panels has been incorporated into the Results section where this figure is discussed.

3. I think Equation S5 in the model is incorrect; it should be $C \rightarrow C+P$, and the model equations and simulations need to be repeated to verify the same conclusions.

We thank the Reviewer for picking up this mistake, which likely occurred when translating the model description into Word. The Supplementary Information has been updated to correct this equation as suggested by the Reviewer. The mathematical model used for the simulations was already correct and therefore our simulations and results were not affected.

4. Please compare the results of the TES to the design in doi.org/10.1093/nar/gky828 where an inhibiting sRNA was used to tune the feedback strength of an autorepressor.

We are grateful to the Reviewer for highlighting this related work and have added a new paragraph to the Discussion to compare these different approaches.

5. Also, please consider how retroactivity could be part of the reason that a difference in fold-change between low and high outputs (lines 178-180) was observed.

It is true that retroactivity could play a role in the changes we observe in the fold-change of the output. Therefore, additional sentences have been added at this point to introduce this concept and explain how and why retroactivity would manifest in this scenario. In addition, a new paragraph has been added to the Discussion covering the role of resource allocation and how the tunable functionality of our devices could help mitigate retroactivity effects. Also, in response to Reviewer 2, we have included an extended mathematical model in Supplementary Text S2 and Figure S2, which captures ribosome allocation dynamics to model explicitly retroactivity effects. These are referenced in the Results section when retroactivity is first raised.

6. The sentences on lines 25 and 110 don't read well; there are also some typos, such as "turner" on line 115, "P_tac" on line 185 etc. Please improve the Figure legends and consider increasing the size of some plots in the figures - for example, the "left and right distributions" in 1D refer to the top, and are indistinguishable in color.

We thank the Reviewer for these suggestions and have reworded the highlighted sentences to clarify our meaning. We have also carefully read through the entire text to try and catch any further minor typos. In regard to the figures, we have checked that all text is 8 pt and restructured several of them to ensure they can be displayed at full size on an A4 page. Previously in the Word document they were shrunk due to the large margins, which would have accounted for the difficulty in reading some of the panels. For the flow cytometry distributions, we have changed the on-state distribution to a solid black outline and the cell autofluorescence distribution to a dashed grey line to help distinguish each more easily. We have also edited all figure captions to improve their clarity where necessary.

Reviewer #2:

Review of “Tunable genetic devices through simultaneous control of transcription and translation” by Vittorio Bartoli, Mario di Bernardo, and Thomas Gorochowski.

This paper proposes a method to tune the (steady state) expression level of a protein, let us say p , as a function of an (inducer) input, let us say u . This tuning is performed by adjusting a parameter which is set by means of an auxiliary (inducer) input, let us say v . Thus, $p = f(u,v)$. The objective is to use v in order to improve the range and switching quality of the function $p = F(u,v)$ as a function of u . A sharp response is particularly desirable for the effective implementation of binary logic gates.

Mathematically, the idea is very simple: u and v induce production of intermediates, and these intermediates can bind to produce a complex; it is this complex that produces the output p . The function $f(u,v)$ saturates when one of u or v is large and the other one is fixed, since there is only so much complex that can be produced by a 1:1 stoichiometry. This explains the graphs that are obtained numerically. (In fact, this explanation predicts a similar plot for p as a function of v , with the roles of input and tuner exchanged.) The results are mathematically very simple and not surprising.

This idea is implemented as follows: the input u controls transcription of an mRNA that cannot be translated unless a specific sRNA (whose transcription is controlled by v) binds to it, thus forming a complex that makes the mRNA's RBS more accessible. The translated protein (say GFP) is the output.

It would appear to me that this idea owes much to ref 6 (ribocomputing devices in paper by groups of Pam silver, Jim Collins, and Peng Yin in Nature 2017, as well as many previous papers by that group). Thus, at first sight, this paper is nice but more at the level of an iGEM project, putting parts together for illustrating a simple mathematical idea. I will defer to experts on mRNA networks to value the novelty in that regard.

We are very grateful to the Reviewer for their careful assessment of our work and valuable comments, but respectfully disagree that our work is at the level of an iGEM team. As also recognized by the other reviewer, the results presented provide a thorough and extensive characterization of a tuning mechanism that goes far beyond the preliminary work that is possible over a summer break. We also would like to stress that in the ribocomputing papers mentioned by the Reviewer, the focus of the authors was purely on the binary on/off behaviors of such RNA devices. Unlike our work, they did not assess the full response dynamics of their devices, overlooked some of the key challenges in their use (e.g. the need for excess trigger RNA), and did not consider their ability to “tune” the response of other types of genetic device (e.g. repressor-based NOT and NOR gates).

The question for me is how important this work is as a general tool in synthetic biology. Much as I like the idea, I have serious doubts in that regard, because I cannot see how this will scale and/or fit in a “modular” design architecture (which the authors use as a motivation).

This work is novel and important in several ways as we have now made clearer in the Introduction and Discussion sections of the revised manuscript. Firstly, it clearly demonstrates how a simple regulatory motif that works at both a transcriptional and translational level can be used to dynamically vary the response dynamics of several common genetic devices in a useful way. This methodology can be easily extended because toe-hold switches can be rationally designed, offering a highly flexible means of tuning the behavior of many other types of genetic device beyond NOT and NOR gates. Secondly, our modelling has elucidated several key design principles for using sRNA

regulation that we experimentally verified. These difficulties are often overlooked when presenting new regulatory mechanisms, which can make their effective application difficult. In terms of scalability, our use of transcriptional signals (promoters) as inputs and outputs means that we can tap into the large array of existing genetic parts and devices that use similar signals. Furthermore, our use of small RNAs and toe-hold switches for tuning places only a minor additional burden on the host cell, ensuring that large circuits can still make effective use of our devices. Regarding modularity, our detailed characterization data was purposefully collected in RPU units, allowing our devices to be directly used in existing genetic circuit automation design tools such as Cello (Nielsen et al. *Science* 352, aac7341, 2016) and combined with the large existing toolkit of logic gates available for this system. Below we comment further on several of the Reviewers specific concerns regarding scalability and modularity.

Issues of retroactivity and resource competition are not addressed, and that's a serious impediment to scale-up of these systems, as discussed for example in [29]. The system, by design, is based on (de-)inhibiting translation. Thus, competition for ribosomes is an especially critical issue. With more gates implemented, this will be a nontrivial issue.

We thank the Reviewer for raising this important point, which was also mentioned by Reviewer 1. To address this, we created a new extended mathematical model of the TES that, as suggested by the Reviewer, captures the competition for ribosomes in the cell using a previously published approach (Gorochowski et al. *ACS Synthetic Biology* 5, 710–720, 2016). Simulations of this model demonstrate that for a sufficiently burdensome output protein, the response functions of the TES would be affected for high 'on' states – similar to that seen in the experiments. However, the changes observed do not account for the increase seen for 'off' states when the tuner activity is increased, suggesting that other factors likely play a role. Discussion about the potential role of retroactivity and relevant citations (Del Vecchio et al. *Molecular Systems Biology* 4, 161, 2008; Del Vecchio & Sontag *European Journal of Control* 3, 389–397, 2009) have been added to the Results section including references to the new model that is presented in Supplementary Text S2 and Figure S2.

Scaling up will also be extremely difficult given the need to have a “tuner” input at each node in a network. (To their credit, the authors are aware of this. Still, it makes the approach very limited.)

It is true that our devices require an additional tuner input and thus some inducible system with a promoter as output is required. However, this does not limit the scalability of these devices in real use-cases. For example, as we currently explain in the Discussion, tunable devices could be selected to be used at specific “sensitive” locations within a circuit to minimize the number of additional inputs required. Or, while the ability to tune devices may be important during the design and optimization process, once a working system is found the inducible tuning promoters could be easily replaced by constitutive ones of an equal strength (effectively removing the external input completely). Furthermore, it is important to recognize that access to inducible systems has grown significantly over recent years with new Marionette strains of *E. coli* (Meyer et al. *Nature Chemical Biology* 15, 196–204, 2018) containing 12 chemically inducible systems that could be immediately used by our devices. That said, the real power of our devices is their ability to adapt to changes dynamically. While for characterization the tuner inputs are externally controlled by inducible promoters, in reality these signals would be driven by endogenous promoters that respond to key cellular signals (e.g. stresses) allowing the devices to adapt their behavior to accommodate physiological changes (Ceroni et al. *Nature Methods* 15, 387–393, 2018). We suspect that in the earlier version of the manuscript we were not clear enough in explaining why the additional tuner inputs are not a significant burden in most

cases and so have expanded the previous description in the Discussion section of the revised manuscript regarding this topic.

The heterogeneity of behavior at the single-cell level (overlap of input-low and input-high distributions) makes the system not very useful, in my opinion, in situations where the wrong “on” vs “off” state is undesirable even if this happens only in a fraction of the population. (For example, if “on” means inducing the release of a toxin, in an engineered therapeutic cell, but a large proportion of cells do so in the wrong environment.) As far as I can see, tuning does not help with this. (Perhaps a refinement of the construct can result in more homogeneity of behavior?)

Single-cell heterogeneity is a major challenge when attempting to control gene expression, and we have been explicit about this difficulty in the original submission. While none of our devices have been able to completely eradicate some intersection between on and off output states, we have been able to show that by combining both the sRNA booster and removal of RiboJ for the TES that a major reduction in intersection from 78% to only 15% for low tuner activities and from 69% to 22% for high tuner activities is possible. This clearly demonstrates that design considerations can affect the performance of our devices in this respect. Furthermore, there are a number of other design improvements that could be explored in the future. For example, new toe-hold switch designs have been released that offer an improved dynamic range and thus separation between off and on output states (Green et al. *Nature* 548, 117–121, 2017). If combined with repressors that display a high transition point (K value) and dynamic range, then it would be expected that full separation could be achieved in the tunable devices. It is worth noting that the new NOR gate we construct (see next comment) performs much better in this regard, showing that large improvements are possible. To ensure the reader is aware of these developments, the Discussion has been updated to discuss the challenges of cell to cell variability and potential future modifications to the design which may elevate its impact on device performance.

The construction of a NOT gate is of some interest, but I believe that the paper would be far more interesting if some nontrivial function, say an AND gate, could be implemented and shown how to tune (with several tuners, if needed).

We agree that the ability to construct more complex tunable logic circuits would increase the impact and usefulness of this work and so have designed and characterized a new 2-input tunable NOR gate. This uses a single tuner to modulate the transition point for both inputs. The major benefit of providing tunable NOT and NOR gates is that they form a functionally complete set of Boolean operators. This means that they can be used to construct all possible combinatorial logic functions. Details of the development of this device and its implications for constructing more complex tunable logic circuits has been added as a new subsection to the Results and a new figure and table has been included to show the design and characterization data (Figure 5; Supplementary Table S3).

Another aspect that could make the paper far more interesting is showing how tuning might be used to diminish fan-out effects. For example, a higher and more homogeneous “high” state might diminish the strength of the “virtual repression” between two competing targets of a given “upstream” (and tunable) module.

This is an interesting idea and the ability for the TES to tune protein synthesis rate would allow for a necessary boost if downstream processes sequestered regulatory molecules (e.g. repressors).

Although experimentally outside the scope of this work, we have added this as a potential application of our devices in the Discussion.

The construct in this paper reminds me of the post-transcriptional regulation in the core fragment / sigma fragment system in Segall-Shapiro et al., Molecular Systems Biology 2014, which also aims at tuning production through a “resource allocator” acting as a tuner (though in a different context, with many targets, and obviously with completely different biological parts, as it has to do with RNAP’s and not sRNA’s and mRNA’s). The authors might want to comment on this analogy.

We appreciate the Reviewer making the connection with the previously developed “resource allocator” and agree that our tunable devices could be used to enable a similar fashion to limit the resources available to a circuit. To elaborate on this idea, some additional sentences have been added to the Discussion and a citation to the Segall-Shapiro et al. paper included.

Some more details on the stochastic modeling could be given, to facilitate reproducibility: how much of each sample path was dropped to assume steady state, how many runs, etc.

As suggested by the Reviewer, further details on how the stochastic modelling was performed have been added to the Materials and Methods section.

I do not understand how normalizing into RPU’s helps when interconnecting systems. While I understand Endy et al.’s reason for normalizing gene expression numbers by means of a comparison with a standard gene expressed in the same temperature, cell, etc., it is unclear to me how using RPU’s helps in ensuring the well-matching of interconnections. A citation or more explanation would be useful.

We apologise for having not made this point clear. The reason for using RPUs is similar to Endy’s, however, the “standard” that RPUs are calculated from is designed to remove potential variation that can occur in the translation step of the fluorescent reporter protein. Specifically, self-cleaving ribozymes (e.g. RiboJ) are used to remove any sequence differences at the 5’-end of a transcript due to the reporter cassette being expressed from a different promoter. These minor differences can cause major changes in the strength of the ribosome binding site controlling translation rate of the reporter protein. By ensuring that precisely the same transcript is produced no matter the promoter used, this effect is mitigated, allowing for a more accurate measurement of relative promoter activity (i.e. transcription rate). This is the reason the term Relative Promoter Units (RPUs) is used. Details regarding the measurement methodology are described in detail in Nielsen et al. *Science* 352, aac7341, 2016.

The reason that adopting RPU units is valuable is that promoters are used in our devices as inputs and outputs. To connect two devices together, we make the output promoter of one device the input promoter of the other. Because we have characterisation data (response functions) for each device in RPUs, we can predict how connected devices will behave and check before assembly that they are compatible (i.e. that the required input range of one device matches the achievable output range of the other). Previous work has demonstrated the power of this methodology in enabling the predictive design of large circuits (Nielsen et al. *Science* 352, aac7341, 2016) and because our characterisation data is measured in RPUs, our devices (NOT and NOR gates) can immediately be used with these existing design tools and part libraries. To ensure the reader is aware of this value, we have added several sentences explaining the above points to the beginning of the Results section when the characterisation approach is introduced.

Lines 25/26: one of the most commonly used is RNA polymerase (RNAP) flux with promoters are used to guide -- I cannot parse this... maybe "in which promoters...?"

This typo has been corrected as suggested by the Reviewer.

Mathematical typo: equation (S12) is missing a term $-kC-C$.

We thank the Reviewer for picking up this mistake, which likely occurred when translating the model into Word. The revised manuscript has been updated as suggested by the Reviewer. The mathematical model used for the simulations included the missing term and so our simulation results were not affected.

Reviewers' Comments:

Reviewer #1:

Remarks to the Author:

Thanks for your reply and for implementing many of my suggestions. The paper is ready for publication.

Reviewer #2:

Remarks to the Author:

The paper has improved considerably, especially given the implementation of the NOR gate.